# The C250T Mutation of *TERTp* Might Grant a Better Prognosis to Glioblastoma by Exerting Less Biological Effect on Telomeres and Chromosomes Than the C228T Mutation

**DOI:** 10.3390/cancers16040735

**Published:** 2024-02-09

**Authors:** Teresa Gorria, Carme Crous, Estela Pineda, Ainhoa Hernandez, Marta Domenech, Carolina Sanz, Pedro Jares, Ana María Muñoz-Mármol, Oriol Arpí-Llucía, Bárbara Melendez, Marta Gut, Anna Esteve, Anna Esteve-Codina, Genis Parra, Francesc Alameda, Cristina Carrato, Iban Aldecoa, Mar Mallo, Nuria de la Iglesia, Carmen Balana

**Affiliations:** 1Medical Oncology, Hospital Clínic, Translational Genomics and Targeted Therapeutics in Solid Tumors, August Pi i Sunyer Biomedical Research Institute (IDIBAPS), 08036 Barcelona, Spain; tgorria@clinic.cat (T.G.); ccrous@clinic.cat (C.C.); epineda@clinic.cat (E.P.); 2Medical Oncology, Institut Catala d’Oncologia (ICO) Badalona, Badalona Applied Research Group in Oncology (B-ARGO Group), Institut Investigació Germans Trias i Pujol (IGTP), 08916 Badalona, Spain; ahernandezg@iconcologia.net (A.H.); mdomenechv@iconcologia.net (M.D.); aesteve@iconcologia.net (A.E.); 3Pathology Department, Hospital Universitari Germans Trias i Pujol, 08916 Badalona, Spain; carosanz.germanstrias@gencat.cat (C.S.); ammunoz.germanstrias@gencat.cat (A.M.M.-M.); ccarrato.germanstrias@gencat.cat (C.C.); 4Department of Pathology, Biomedical Diagnostic Centre (CDB) and Neurological Tissue Bank of the Biobank-IDIBAPS, Hospital Clinic, University of Barcelona, 08036 Barcelona, Spain; pjares@clinic.cat (P.J.); ialdecoa@clinic.cat (I.A.); 5Cancer Research Program, Institut Hospital del Mar d’Investigacions Mèdiques (IMIM), 08003 Barcelona, Spain; oarpi@imim.es; 6Molecular Pathology Research Unit, Hospital Universitario de Toledo, 45007 Toledo, Spain; bmelendez@sescam.jccm.es; 7Centro Nacional de Análisis Genómico, C/Baldiri Reixac 4, 08028 Barcelona, Spain; marta.gut@cnag.eu (M.G.); anna.esteve@cnag.eu (A.E.-C.); genis.parra@cnag.eu (G.P.); 8Badalona Applied Research Group in Oncology (B-ARGO Group), Institut Investigació Germans Trias i Pujol (IGTP), 08916 Badalona, Spain; 9Pathology Department, Neuropathology Unit, Hospital del Mar, Institut Hospital del Mar d’Investigacions Mèdiques (IMIM), 08003 Barcelona, Spain; 11669faq@gmail.com; 10Unidad de Microarrays, Institut de Recerca Contra la Leucèmia Josep Carreras (IJC), ICO-Hospital Germans Trias i Pujol, Universitat Autònoma de Barcelona, 08916 Badalona, Spain; mmallo@carrerasresearch.org; 11IrsiCaixa AIDS Research Institute, Hospital Universitari Germans Trias i Pujol, 08916 Badalona, Spain; ndelaiglesia@irsicaixa.es

**Keywords:** *TERTp* mutations, glioblastoma, *MGMTp* methylation, RNA-Seq differential expression analysis, whole-exome sequencing

## Abstract

**Simple Summary:**

In our glioblastoma patients treated with standard therapy, the *TERTp* C250T mutation occurred less frequently than the C228T mutation. Patients with the C250T mutation had better prognosis than those with either *TERTp*-wt or *TERTp* C228T mutations, even when adjusted for key glioblastoma prognostic factors. This may be due to the lesser involvement of the C250T mutation in telomere- and chromosome-related pathways, as evidenced by the results of a gene enrichment analysis adjusted for *MGMTp* methylation status: *TERTp* C250T was differentially enriched compared to *TERTp*-wt and C228T. There were no differences according to *TERTp* mutation status in the mutations or copy number variants of other genes commonly present in glioblastoma. The biological pathways by which *TERTp* and *MGMTp* exert their effects are independent.

**Abstract:**

The aim of this study was to determine how *TERTp* mutations impact glioblastoma prognosis. Materials and Methods: *TERTp* mutations were assessed in a retrospective cohort of 258 uniformly treated glioblastoma patients. RNA-sequencing and whole exome sequencing results were available in a subset of patients. Results: Overall, there were no differences in outcomes between patients with mutated *TERTp*-wt or *TERTp*. However, we found significant differences according to the type of *TERTp* mutation. Progression-free survival (mPFS) was 9.1 months for those with the C250T mutation and 7 months for those with either the C228T mutation or *TERTp*-wt (*p* = 0.016). Overall survival (mOS) was 21.9 and 15 months, respectively (*p* = 0.026). This differential effect was more pronounced in patients with *MGMTp* methylation (mPFS: *p* = 0.008; mOS: *p* = 0.021). Multivariate analysis identified the C250T mutation as an independent prognostic factor for longer mOS (HR 0.69; *p* = 0.044). We found no differences according to *TERTp* mutation status in molecular alterations common in glioblastoma, nor in copy number variants in genes related to alternative lengthening of telomeres. Nevertheless, in the gene enrichment analysis adjusted for *MGMTp* methylation status, some Reactome gene sets were differentially enriched, suggesting that the C250T mutation may exert a lesser effect on telomeres or chromosomes. Conclusions: In our series, patients exhibiting the C250T mutation had a more favorable prognosis compared to those with either *TERPp*-wt or *TERTp* C228T mutations. Additionally, our findings suggest a reduced involvement of the C250T mutation in the underlying biological mechanisms related to telomeres.

## 1. Introduction

Glioblastoma is currently defined by the WHO 2021 classification as the highest tumor grade in the astrocyte lineage, without mutations in the isocitrate dehydrogenase (*IDH*) genes. It is the most frequent malignant CNS tumor in adults, with few therapeutic options, and progress in the search for new treatment strategies has been very slow. Median survival for patients receiving the standard treatment of maximal resection followed by post-operative chemoradiotherapy with temozolomide is around 15 months, while it is only 8 months for those who are ineligible for this treatment [1]. Recognized prognostic factors used for therapeutic decisions include age, extent of surgery, KPS, and methyl-guanine-methyl-transferase promoter (*MGMTp*) methylation [2,3].

Telomeres are highly repetitive non-coding DNA regions located at the terminal end of eukaryotic chromosomes; they prevent chromosome recombination, end-to-end fusion, and DNA-damage recognition. They control the replicative capacity of human cells, as they shorten with each cell division until they reach a certain limit, after which they cannot continue to function, leading to cell apoptosis or senescence [4]. Telomerase reactivation may occur via multiple genetic and epigenetic mechanisms, including *TERT* and *TERC* (the RNA template component of the telomerase structure) amplification; genomic rearrangements; somatic mutations within *TERTp*; and epigenetic modifications through *TERTp* methylation [4,5,6,7]. In addition, telomeres may be maintained by alternative lengthening of telomeres (ALT), which involves mutations in the genes encoding for the α-thalassemia/mental retardation syndrome X-linked protein (ATRX); the death domain-associated protein (DAXX); and the SWI/SNF-related, matrix-associated, actin-dependent regulator of chromatin, subfamily A, member 1 (SMARCA1) [8,9]. Telomerase activation is estimated to occur in 85–90% of cancers; ALT pathway is observed in approximately 10–15% of them [10].

Telomerase reverse transcriptase promoter (*TERTp*, HGNC:11730) mutations are frequent in multiple tumor types [11], and are present in 70–80% of glioblastomas and around 80% of oligodendrogliomas, where they always co-occur with *IDH1/2* mutations [11,12,13]. The presence of a *TERTp* mutation in a morphologically low-grade *IDH*-wild-type (wt) glioma reclassifies it as glioblastoma, as these tumors will behave similarly to glioblastoma in terms of outcome [14]. However, *TERTp* mutations do not constitute a molecular grading factor among oligodendrogliomas, which have a much better prognosis [15,16]. Several *TERTp* mutations have been described but they primarily occur at one of two hotspots as a C-to-T transition in the coding strand. The most frequent transitions—g.1295228C>T (chr5, hg19) (C228T) and g.1295250C>T (chr5, hg19) (C250T)—are located at −124 and −146 bp, respectively, upstream of the start codon, and are mutually exclusive [11]. C228T is by far the most common *TERTp* mutation except in skin cancer, where C250T occurs almost as frequently [17]. 

Contradictory results have been reported on the prognostic impact of *TERTp* mutations in glioblastoma. Some studies have found that *TERTp* mutations are a negative prognostic factor, both in gliomas in general [18] and in glioblastomas in particular [12,19,20,21], while others have found no differences in outcomes associated with *TERTp* mutations [22,23,24,25,26] if other known prognostic factors are taken into account [22]. The potential interaction between *MGMTp* methylation and *TERTp* mutations on clinical results has also been extensively investigated, though without definitive results at the biological level [22,25,26,27,28].

We have analyzed the prognostic value of *TERTp* mutations in a retrospective cohort of 258 uniformly treated glioblastoma patients. In addition to data on *TERTp* mutation status, we have collected information on all other known prognostic factors, including *MGMTp* methylation status. For further analyses of other molecular alterations potentially related to *TERTp* mutations and their effect on prognosis, we had RNA-sequencing (RNA-Seq) and whole exome sequencing analysis (WES) results available in a subset of the patients.

## 2. Material and Methods

### 2.1. Patients

This was a retrospective multicenter study including 258 patients diagnosed and treated at six Spanish institutions from May 2005 to November 2022. All patients were diagnosed with glioblastoma *IDH* wt and were uniformly treated with the standard treatment of surgery followed by concomitant radiation plus temozolomidec, and adjuvant temozolomide. The patients were included in two previous studies by the GLIOCAT group. Clinical data were collected from hospital records. The primary endpoint of the study was to determine the impact of *TERTp* mutation status on patient outcome, taking into account other known prognostic markers.

### 2.2. Molecular Analyses

Diffuse midline glioma, H3 p.K28M (K27M)/p.K28I (K27I) and diffuse hemispheric glioma H3.3 p.G35R (G34R)/p.G35V (G34V) were ruled out via immunohistochemical analysis as previously described [29] and WES. *MGMTp* methylation was assessed via methylation-specific PCR as previously described [30]. The presence of *IDH* mutations was ruled out via immunohistochemistry, and/or via PCR in patients younger than 56 years or via WES. *TERTp* mutation status was determined using PCR/Sanger sequencing in 240 patients (93%); the Oncomine Comprehensive Assay and the IonChef-S5 Sequence Platform (Thermo Fisher, Waltham, MA, USA) followed by analysis with Ion Reporter Software (Thermo Fisher, https://ionreporter.thermofisher.com/ir/, last accessed on 21 November 2022) in 12 (4.7%); and with FoundationOne CDx (Roche Foundation, Cambridge, MA, USA) in 6 (2.3%). The lack of detection of *TERTp* mutations on these last two tests did not automatically classify the tumor as *TERTp*-wt unless the wt status was confirmed through alternative methods.

For PCR/Sanger sequencing, *TERTp* was amplified with the forward 5′-GCACCCGTCCTGCCCCTTCACC-3′ and reverse 5′-GGCTTCCCACGTGCGCAGCAGGA-3′ primers using DNA obtained from macrodissected formalin-fixed paraffin-embedded (FFPE) tumor tissue. The PCR conditions were standard, with an annealing temperature of 62 °C, 2 mM MgCl_2_ and 5% DMSO. PCR bands were cleaned up with Ilustra^TM^ExoProStar^TM^ 1-Step (GE HealthCare, Chicago, IL, USA) and sequenced with the forward primer and the BigDye^®^ Terminator v1.1 Cycle sequencing kit (Applied Biosystems, Waltham, MA, USA). Sequences were processed on SeqStudio Genetic Analyzer (Applied Biosystems) and analyzed with SeqScanner2 software 2 (version 2.0).

Results of the *TERTp* mutation analyses were unified, and tumors were classified as having the C228T or the C250T mutation or as *TERTp*-wt.

RNA-Seq had been performed on FFPE samples from 85 patients, as previously described [31]. Differential gene expression analysis comparing samples with the C228T or C250T mutation, or *TERTp*-wt, was performed with limma-voom [32] using the RSEM [33] raw counts as input and adjusting the model for age, sex and, in a second analysis, also by *MGMTp* methylation status. Data were analyzed with the Kruskal–Wallis test with *TERT* log2cpm. Gene set enrichment analysis (GSEA) was done with the R fgsea package [34,35], using the output of the differential expression analysis. Genes were ranked by the limma-moderated t-statistic (Preranked-GSEA) and Reactome gene sets were used as the predefined gene list. The direction of the enrichment was provided by the normalized enrichment score (NES), a metric that corrects for differences in the enrichment score between gene sets due to differences in gene set size. Reactome pathways with an adjusted *p*-value < 0.05 were considered significant.

WES had been performed on DNA from the FFPE samples from 92 patients. SureSelect Human All Exon V5 (Agilent Technologies, Santa Clara, CA, USA) capture was used for whole exome enrichment with the modified protocol for FFPE samples. The KAPA Hyper Prep Kit (Roche, Basel, Switzerland) was used for DNA pre-capture library preparation. The starting material was 0.2–0.5 μg of FFPE-extracted genomic DNA (gDNA). The gDNA was sheared on a Covaris E220 Plus (Covaris, Woburn, MA, USA), end-repaired and adenylated. Illumina platform-compatible adaptors with unique dual indexes and unique molecular identifiers (Integrated DNA Technologies, Coralville, IA, USA) were ligated. The adaptor-modified end library was amplified with 8–18 PCR cycles using the KAPA HiFi HotStart ReadyMix (2X) BARBB (Roche, Basel, Switzerland), varying in function of the initial FFPE gDNA quality metrics. The quality of the PCR product was controlled on the Agilent 2100 Bioanalyzer 7500 chip (Agilent Technologies) to confirm size range and quantity of the library. It was then hybridized for 24 h at 65 °C with the 2720 Thermal Cycler (Applied Biosystems, Thermo Fisher), The hybridization mix was washed and the eluate PCR-amplified for 12 cycles using KAPA HiFi HotStart ReadyMix (2X). The final library size and the concentration were determined on the 2100 Bioanalyzer 7500 (Agilent Technologies). 

The captured libraries were sequenced on NovaSeq 6000 (Illumina, San Diego, CA, USA) in paired-end mode with a read length of 2 × 101 bp following the manufacturer’s protocol for dual indexing. Image analysis, base calling and quality scoring of the run were processed using the manufacturer’s software, Real Time Analysis (RTA 3.4.4), followed by generation of FASTQ sequence files.

After sequencing, reads were mapped to the human genome (hs37d5) using the BWA-mem [36] with default parameters. Alignment files (BAM format) containing only properly paired, unique mapping reads were processed using Picard tools version 1.110 (http://broadinstitute.github.io/picard/ (accessed on 31 August 2017)) to add read groups and remove duplicates. The Genome Analysis Tool Kit was used for local indel realignment and base recalibration (https://www.broadinstitute.org/gatk/about/citing (accessed on 31 August 2017)) [37].

Processed BAM files were used to perform somatic variant calling of single nucleotide variants, and small insertions and deletions with Mutect2 (GATK v4.0.8.1). Since matched control samples were not available, Mutect2 was run in tumor-only mode. To further discriminate somatic from germline variants, two sets of variants were provided to Mutect2: (1) the aggregate variants from a “panel of normals” of 400 individuals; and (2) a set of human variants from gnomAD (https://gnomad.broadinstitute.org (accessed on 31 August 2017)), as included in the GATK suite. Only the variants considered as PASS by Mutect2, using FilterMutectCalls (GATK v4.0.8.1), were considered for the study. To further eliminate potential artefacts, variants had to have a minimum depth of coverage of ten reads, at least three reads supporting the alternative allele, and an alternative allele frequency >0.05.

Functional annotations were added to the resulting VCF file using snpEff [38] with the gene annotation obtained from ENSEMBL version 75 (http://www.ensembl.org/ (accessed on 31 August 2017)).

Copy number variants (CNV) were predicted using Control-FREEC [39]. A baseline created with the panel of normals was used as a control. 

### 2.3. Statistical Analyses

Categorical variables were compared with the χ^2^ or Fisher’s exact test, as appropriate, and continuous variables with the Kruskal–Wallis test. Progression free survival (mPFS) was calculated from the date of surgery to the date of progression, death, last follow-up, or administrative censoring (15 March 2023). Overall survival (mOS) was calculated from the date of surgery to the date of death from any cause, last follow-up, or administrative censoring. Median PFS and OS, with their 95% confidence intervals (CIs), were estimated with the Kaplan–Meier method and compared with the log-rank test. Multivariate Cox proportional hazards models were used to assess the prognostic value of *TERTp*, adjusted by *MGMTp* methylation status, KPS, age and extent of surgery. Hazard ratios (HRs) and their 95% CIs were calculated. All reported *p*-values were two-sided. Analyses were performed with R software v3.4.2 and SPSS v24 (IBM).

### 2.4. Ethics Statement

This study was approved by the Institutional Review Board of the Hospital Germans Trias i Pujol (PI-14-016 and PI-18-259), and by the Ethics Committees of all the participating institutions and their biobanks. The study was conducted in accordance with the ethical standards as laid down in the 1964 Declaration of Helsinki and its later amendments. All patients or their representatives gave their written informed consent to participate in this study.

## 3. Results

### 3.1. Patients

Patient characteristics are shown in Table 1. Ninety patients were over 65 years of age. Surgery consisted of gross total resection in 103 patients, partial resection in 122, and biopsy only in 32. KPS was <80% in 59 patients. *MGMTp* methylation status was known in 253 patients (98.4%), 135 (52.3%) of whom had *MGMTp* methylation. At recurrence, patients were treated with second surgery, re-irradiation, or systemic treatment (including nitrosoureas or temozolomide), or were included in clinical trials. Bevacizumab was administered to 156 patients (60.7%) as a second or further line of treatment.

### 3.2. TERTp Mutations

*TERTp* mutations were detected in 202 patients (78.7%). The C228T mutation was detected in 56.2% of patients and the C250T mutation in 22.1%, while 21.3% had *TERTp*-wt. In addition, one patient had the C229T mutation but was excluded from the analysis. This patient was a 53-year-old woman with a gross total resection and no *MGMTp* methylation; she was progression free for 11.2 months and survived for 19.7 months (Table 1).

### 3.3. Outcomes

Progression or death occurred in 229 patients (89.1%); 107 (76.7%) had died at the time of analysis. Median PFS for all patients was 8 months (95% CI 7.4–91) and mOS was 16.3 months (95% CI 14.2–18).

Age ≤ 65 years, KPS ≥ 80%, gross total resection, and *MGMTp* methylation improved prognosis in the univariate analysis. No significant differences in mPFS or mOS were observed according to *TERTp* mutation status (Table 2 and Appendix A). However, we found significant differences in both mPFS and mOS according to the type of *TERTp* mutation. Median OS was 21.9 months (95% CI 15.1–32.7) for patients with the C250T mutation, compared to 16 months (95% CI 14–17.1) for those with the C228T mutation, and 14.2 months (95% CI 11.9–22.4) for those with *TERTp*-wt (*p* = 0.047) (Table 2 and Appendix A). When patients with the C250T mutation were compared to all other patients (C228T and *TERTp*-wt), the significance increased: mOS for those with the C250T mutation was 21.9 months (95% CI 15.1–32.7), while it was 15 months (95% CI 14–17.1) for those with either the C228T mutation or *TERTp*-wt (*p* = 0.016) (Table 2 and Appendix A). Differences in mPFS were similar (Table 2 and Appendix A).

As expected, patients with methylated *MGMTp* had better prognosis than those with unmethylated *MGMTp*, regardless of *TERTp* mutation status (Appendix A). However, the greatest benefit in both mPFS and mOS was for patients with both *MGMTp* methylation and the *TERTp* C250T mutation: mPFS for these patients was 12.1 months (95% CI 9.8–22.3) (*p* = 0.002) and mOS was 24.8 months (95% CI 21.7–not reached) (*p* = 0.007) (Table 2 and Figure 1A,B).

**Table 2 cancers-16-00735-t002:** Median progression-free survival (mPFS) and median overall survival (mOS) according to *TERTp* mutation status and *MGMTp* methylation status.

Comparisons		mPFS		mOS	
		Months (95% CI)	*p*	Months (95% CI)	*p*
All patients		8 (7.4–9.1)		16.3 (14.2–18)	
*TERTp* wt vs. mut	wt	7 (6.3–9)	0.103	14.2 (11.9–22.4)	0.801
	C228T + C250T	8.4 (7.6–9.8)		16.7 (14.7–18.4)	
*TERTp* C250T vs. C228T vs. wt	C250T	9.1 (7.8–12.8)	0.048	21.9 (15.1–32.7)	0.047
	C228T	8.1 (7.3–9.4)		16 (14–17.1)	
	wt	7 (6.3–9)		14.2 (11.9–22.4)	
*TERTp* C250T vs. Wt + C228T	C250T	9.1 (7.8–12.8)	0.026	21.9 (15.1–32.7)	0.016
	wt + C228T	7.7 (7.1–8.9)		15 (14–17.1)	
*MGMTp* status & *TERTp* C250T vs. Wt + C228T	Met & C250T	12.1 (9.8–22.3)	0.002	24.8 (21.7–NR)	0.007
	Met & wt + C228T	8.1 (7.3–1.3)		14.2 (13.4–2.7)	
	UnMet & wt + C228T	7.1 (6.7–9.2)		15 (14–17.1)	
	UnMet & C250T	7.9 (6.6–13)		18 (13.2–32.7)	

wt, wild-type; mut, mutated; Met, methylated; UnMet, unmethylated; NR: not reached; When *TERTp* mutation status was included in multivariate analyses together with age, extent of surgery, KPS and *MGMTp* methylation, the C250T mutation—but not the C228T mutation—emerged as a prognostic factor for longer mOS (HR = 0.69, 95% CI 0.48–0.99; *p* = 0.04) (Table 2 and Figure 2A,B).

**Figure 2 cancers-16-00735-f002:**
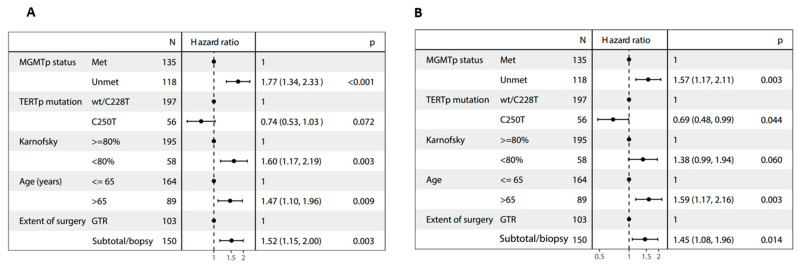
Forest plots of multivariate analyses of (**A**) progression-free survival and (**B**) overall survival.

Of the 92 patients with informative WES results, three harbored *TERT* variants, all of which were protein-coding missense mutations: one with c.2935C>T; one with c.2572C>T; and one with both c.1457G>A and c.470C>T. The first three of these mutations are possibly pathogenic, while the fourth is considered benign. All three patients also had the *TERTp* C228T mutation. We detected no CNV of the *TERT* gene.

The analysis of genes related to glioblastoma, ALT or *TERT* transcription: *EGFR* amplification; mutations in *EGFR*, *TP53*, *PTEN*, *BRAF*, *PI3K*, *MYC*, *DAXX*, *SMARCA*, and *ATRX*; CNV of *CDKN2A/B*; and RNA expression of the LncRNA *TERC* showed no differences in the distribution of these alterations depending on the *TERTp* mutation status (Table 3). Similarly, there were no significant distinctions when patients were grouped based on C250 or C228T mutations and wild-type status. 

Additionally, there were no fusions of another gene with *TERT* and no differences in *TERT* RNA expression associated with *TERTp* mutation status (Appendix A). 

RNA-Seq comparing samples according to *TERTp* mutation status, using a false discovery rate (FDR) threshold of <5%, did not reveal any significant differences associated with *TERTp* mutation status. However, the gene set enrichment analysis showed that some Reactome gene sets were differentially enriched, with significant differences in the NES (Figure 3). These results were maintained even when adjusting for *MGMTp* methylation status. Compared to *TERTp*-wt samples, those with *TERTp* mutations were enriched in pathways involved with telomeres (packaging of telomere ends; telomere maintenance; inhibition of DNA recombination at telomere pathways; DNA damage/telomere stress-induced senescence; and extension of telomeres) and chromosomes (prophase/prometaphase chromosome condensation; chromosome maintenance; chromatin-modifying enzymes; and chromatin organization). 

Interestingly, the greatest enrichment of these pathways was seen in the comparison between samples with the C228T mutation and those with *TERTp*-wt, particularly in pathways involved in prophase/prometaphase chromosome condensation. There was also some enrichment observed in the comparison between samples with the C250T mutation and those with *TERTp*-wt, but the significance was maintained only in pathways involved in telomere maintenance; prophase/metaphase chromosome condensation; chromosome maintenance; chromatin-modifying enzymes; and chromatin organization. No significant differences in these pathways were found between samples with the C228T or C250T mutations.

## 4. Discussion

In the present study, *TERTp* mutations in general did not affect prognosis, but the *TERTp* C250T mutation was associated with longer mPFS (*p* = 0.026) and mOS (*p* = 0.016) compared to the C228T mutation or *TERTp*-wt. The observed effect persisted in terms of mOS during multivariate analyses, which incorporated established prognostic factors such as age, extent of surgery, KPS, and *MGMTp* methylation status (*p* = 0.04). Specifically, the C250T mutation exhibited a favorable impact on prognosis, even among patients with *MGMTp* unmethylation. The mOS was 24.8 months for individuals with *MGMTp* methylation and 18 months for those without *MGMTp* methylation, surpassing the outcomes associated with the C228T mutation (16 and 14.8 months, respectively).

Notably, *TERTp* mutations, particularly C228T, have been linked to poorer prognosis in head and neck cancer patients [40]. Conversely, a study involving 887 gliomas of various grades reported a positive association between the C250T mutation and improved prognosis [41].

The adverse prognostic implications of *TERTp* mutations in lower-grade astrocytomas prompted numerous investigations into their potential impact on glioblastoma prognosis, particularly their potential interaction with *MGMTp* methylation [19,22,25,26,27,42]. However, inconsistencies arose from variations in study methodologies, including the incorporation of recognized prognostic factors, diverse treatment approaches, and limited examination of differences in prognosis based on the specific type of *TERTp* mutation.

An analysis by Nguyen et al., involving 303 glioblastoma patients, failed to identify significant differences in mPFS or mOS based on *TERTp* mutation status. Interestingly, they observed that *MGMTp* methylation only improved prognosis in the presence of *TERTp* mutations, while the mutations worsened prognosis in patients without *MGMTp* methylation [25]. A separate study by Arita et al., encompassing 260 *IDH*-wt glioblastoma patients, validated in a larger cohort, suggested that patients with both *MGMTp* methylation and *TERTp* mutations exhibited the most favorable prognosis, followed by those with *TERTp* wt and *MGMTp* methylation [22].

Despite these findings, data on the specific impact of the type of *TERTp* mutation were lacking. Furthermore, a meta-analysis indicated that the influence of *TERTp* mutations might be modulated by *MGMTp* methylation, suggesting that not all patients with methylated *MGMTp* would benefit from temozolomide, but only those with concurrent *TERTp* mutations [43].

Supporting this notion, a methylome analysis incorporating *TERTp* mutation status revealed a distinct methylation profile between mutated and wild-type *TERTp* tumors, implying potential differences in sensitivity to temozolomide [28].

In contrast, other studies have reported a lack of interaction between *MGMTp* methylation and *TERTp* mutation status [26,27]. Gramatzki et al. [26] included 298 patients from the German Glioma Network (GGN), 205 of whom received the standard treatment, and 302 patients from an independent retrospective cohort, 238 of whom received the standard treatment. Each cohort was analyzed separately, and neither showed variations in mPFS or mOS based on the combination of *MGMTp* methylation and *TERTp* mutations. In the GGN cohort, there were 42 patients with the C250T mutation, while the retrospective cohort included 45 patients with this mutation. Although slight variations in mPFS and mOS were observed initially between these two mutations, these distinctions ceased to hold statistical significance in the subsequent multivariate analyses (supplementary data in [26]).

Our findings in the present study indicate that both *MGMTp* methylation and *TERTp* mutations seem to affect prognosis independently, and that the *TERTp* C250T mutation improved the prognosis of patients regardless of their *MGMTp* methylation status. 

The differential prognosis based on *TERTp* mutational status observed within our own patient cohort could stem from genomic disparities linked to *TERT* RNA transcription and ALT, both of which have been associated with *TERTp* mutation status [11]. Transcription factors are essential both to the regulation of *TERT* transcription and to the accessibility of DNA [44]. Somatic *TERTp* mutations have been shown to create an E-twenty-six (ETS) transcription factor binding site, enhancing the transcriptional activity of *TERTp* [17,45].

These mutations are also associated with increased telomerase activity and *TERT* upregulation in gliomas. [11,46] Specifically, gliomas with the C228T mutation exhibited a 14-fold increase in mRNA expression compared to *TERTp*-wild-type tumors, while those with the C250T mutation had a 7-fold increase (*p* < 0.001). This differential expression was hypothesized to result from varying access to transcription factors due to chromatin remodeling [24]. Notably, while both mutations generate the ETS binding site, only the C250T mutation appears to be influenced by non-canonical NF-kappa B signaling, and ETS binding to the mutant *TERTp* may not be sufficient to drive its transcription [47].

Contrary to these previous findings, our study did not identify differences in *TERT* expression based on *TERTp* mutation status. Several factors could contribute to this lack of association, including the small sample size; potential variations in tumor representation among samples; different combinations of *TERT* alpha and beta transcripts [48,49]; and the possibility of alternative biological mechanisms influencing *TE*RT expression. The study by Salgado et al. [50] in melanoma, which similarly failed to establish a clear association between *TERT* expression levels and mutational status, emphasized the role of promoter methylation in *TERT* expression. This underscores the complexity of regulatory mechanisms governing *TERT* expression and highlights the need for further exploration to unravel these intricate relationships.

Nevertheless, in contrast to *TERTp*-wt tumors, we found that those with *TERTp* mutations had enrichment of the pathways involved in the biological mechanisms of telomeres and chromosomes. Differences between mutated and wt tumors were highly significant, and we observed more differential enrichment between tumors with the C228T mutation and *TERTp*-wt tumors than between those with the C250T mutation and *TERTp*-wt tumors. This leads us to suggest that the C250T mutation is less efficient in activating telomere lengthening than the C228T, which could explain both the deleterious impact of C228T and the beneficial impact of the C250T mutation on patient prognosis. These differences were maintained even when adjusted for *MGMTp* methylation status, suggesting that the effect of *TERTp* mutations and *MGMTp* methylation follow different—and not necessarily related—biological pathways, so that their effect on prognosis could be additive or restrictive.

In addition, we found that mutations in genes involved in ALT (*ATRX*, *DAXX* and *SMARCA1*), or in those indirectly regulating *TERT* transcription—including the *MYC* oncogene, which acts through the Myc/Max/Mad protein family—and the *PIK3* gene family [9,51,52], were not universally or exclusively present in *TERTp*-wt tumors, which leads us to surmise that these tumors do not have telomeres as their main oncogenic pathway [53]. There were no rearrangements of *TERT* in our series, and the only three patients with *TERT* mutations also had the *TERTp* C228T mutation. Finally, our study did not reveal any special enrichment of a specific pathway in *TERTp*-wt patients. Consequently, we cannot posit this as an alternative explanation of our findings regarding the impact of *TERTp* mutations on outcome.

Our study has several limitations, including its retrospective nature and the relatively low number of patients with tumors with *TERTp*-wt or the C250T mutation, which is a characteristic shared by several previous reports [22,25,26]. In addition, we had no data on *TERTp* methylation and no functional data on the effect of differences in genomic expression on telomere lengthening or telomerase levels in tumors with different *TERTp* mutational status. This potential interaction between genomic expression and *TERTp* mutations warrants further investigation in future studies, especially in trials of *TERT*-targeted therapies.

## 5. Conclusions

In contrast to other studies, our univariate analyses showed a significant association between the C250T mutation and better prognosis, which remained significant in the multivariate analyses even when other prognostic factors were included. These differences with previous studies could be due to the highly curated clinical data we collected, as well as to the fact that we included a homogeneously treated series of patients.

In summary, our findings strongly indicate that patients carrying the *TERTp* C250T mutation exhibit a more favorable prognosis compared to their counterparts, irrespective of their *MGMTp* methylation status. This improved prognosis associated with the C250T mutation may be attributed to its lesser involvement in telomere activity compared to the C228T mutation, thereby limiting its potentially deleterious effects on patient outcomes.

Notably, our investigation did not reveal significant differences in molecular alterations related to other glioblastoma-associated genes, or in genes associated with the ALT system. This suggests that the prognostic impact of the *TERTp* C250T mutation may be specifically linked to its modulation of telomere activity rather than broader alterations in the molecular landscape of glioblastomas. Further research and comprehensive analyses are warranted to delve into the intricate molecular mechanisms underlying the observed prognostic differences associated with *TERTp* mutations, and to unravel potential therapeutic implications.

## Figures and Tables

**Figure 1 cancers-16-00735-f001:**
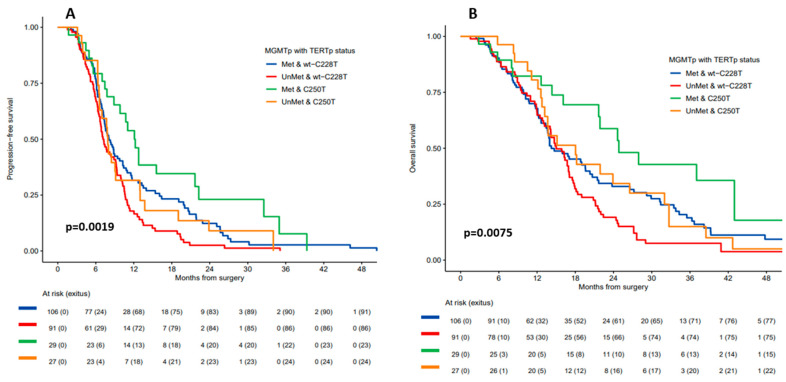
(**A**) Progression-free survival and (**B**) overall survival according to *MGMTp* methylation status and *TERTp* mutation status (C250T versus C228T mutation + *TERTp*-wt).

**Figure 3 cancers-16-00735-f003:**
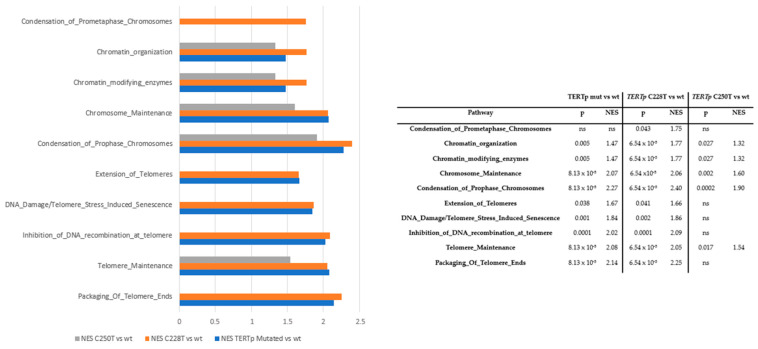
Gene enrichment analysis according to *TERTp* mutation status adjusted for age, sex and *MGMTp* methylation status. NES: normalized enrichment score.

**Table 1 cancers-16-00735-t001:** Patient characteristics.

Characteristic	All Patients *n* = 257 ^a^	*TERTp* C228T *n* = 145	*TERTp* C250T *n* = 57	*TERTp* Wild-Type *n* = 55	*p * ^b^
Sex					0.787
Male	163 (63.4%)	93 (64.1%)	34 (59.6%)	36 (65.5%)	
Female	94 (36.6%)	52 (35.9%)	23 (40.4%)	19 (34.5%)	
Age, yrs—median (range)	60.6 (19–81)	61.5 (31–81)	61.5 (32–81)	57.2 (19–78)	0.052
≤65	167 (65.0%)	89 (61.4%)	35 (61.4%)	43 (78.2%)	0.069
>65	90 (35.0%)	56 (38.6%)	22 (38.6%)	12 (21.8%)	
KPS					0.344
≥80%	198 (77.0%)	109 (75.2%)	48 (84.2%)	41 (74.5%)	
<80%	59 (23.0%)	36 (24.8%)	9 (15.8%)	14 (25.5%)	
Extent of surgery					0.200
Gross total resection	103 (40.1%)	57 (39.3%)	28 (49.1%)	18 (32.7%)	
Subtotal/biopsy	154 (59.9%)	88 (60.7%)	29 (50.9%)	37 (67.3%)	
*MGMTp* status					0.686
Methylated	135 (52.5%)	81 (55.9%)	29 (50.9%)	25 (45.5%)	
Unmethylated	118 (45.9%)	62 (42.8%)	27 (47.4%)	29 (52.7%)	
Unknown	4 (1.6%)	2 (1.4%)	1 (1.8%)	1 (1.8%)	

^a^ Of the original 258, one had the C229T mutation but was not included in the prognostic analysis. This patient was a 53-year-old woman with a gross total resection and no *MGMTp* methylation; she was progression free for 11.2 months and survived for 19.7 months. ^b^ Shows *p*-values for the comparison of the three *TERTp* groups (C228T vs. C250T vs. wild-type).

**Table 3 cancers-16-00735-t003:** Molecular alterations according to *TERTp* mutation status, as detected via RNA-Seq and/or WES in tumor samples.

Molecular Alteration	Status	All Patients *n* = 92	*TERTp* C228T *n* = 57	*TERTp* C250T *n* = 18	*TERTp*-wt *n* = 17	*p * ^a^
*EGFR* mutation	No	79 (85.9%)	50 (87.7%)	13 (72.2%)	16 (94.1%)	0.216
	Yes	13 (14.1%)	7 (12.3%)	5 (27.8%)	1 (5.88%)	
*EGFR* amplification	No	49 (53.3%)	27 (47.4%)	12 (66.7%)	10 (58.8%)	0.316
	Yes	43 (46.7%)	30 (52.6%)	6 (33.3%)	7 (41.2%)	
*P53* mutation	No	73 (79.3%)	47 (82.5%)	13 (72.2%)	13 (76.5%)	0.586
	Yes	19 (20.7%)	10 (17.5%)	5 (27.8%)	4 (23.5%)	
*PTEN* mutation	No	63 (68.5%)	40 (70.2)	11 (61.1%)	12 (70.6%)	0.754
	Yes	29 (31.5%)	17 (29.8)	7 (38.9)	5 (29.4%)	
*BRAF* mutation	No	89 (96.7%)	57 (100%)	16 (88.9%)	16 (94.1%)	0.052
	Yes	3 (3.26%)	0 (0.00%)	2 (11.1%)	1 (5.88%)	
*CDKN2AB* Loss	No loss	41 (44.6%)	26 (45.6%)	8 (44.4%)	7 (41.2%)	0.949
	Loss	51 (55.4%)	31 (54.4%)	10 (55.6%)	10 (58.8%)	
*PIK3* family mutation	No	60 (65.2%)	39 (68.4%)	10 (55.6%)	11 (64.7%)	0.125
	Yes	32 (34.8%)	18 (31.6%)	8 (44.4%)	6 (35.3%)	
*MYC* mutation	No	89 (96.7%)	56 (98.2%)	18 (100%)	15 (88.2%)	0.114
	Yes	3 (3.26%)	1 (1.75%)	0 (0.00%)	2 (11.8%)	
*DAXX* mutation	No	90 (97.8%)	56 (98.2%)	17 (94.4%)	17 (100%)	0.619
	Yes	2 (2.17%)	1 (1.75%)	1 (5.56%)	0 (0.00%)	
*SMARCA* family mutation	No	77 (83.7)	49 (86.0%)	14 (77.8%)	14 (82.4)	0.249
	Yes	15 (16.3)	8 (14.0)	4 (22.2)	3 (17.6)	
*ATRX* mutation	No	86 (93.5%)	52 (91.2%)	17 (94.4%)	17 (100%)	0.724
	Yes	6 (6.52%)	5 (8.77%)	1 (5.56%)	0 (0.00%)	
*TERC* LncRNA	Differential expression	No differences in the expression of *TERC* LncRNA				

^a^ Shows *p*-values for the comparison of the three *TERTp* groups (C228T vs. C250T vs. wild-type).

## Data Availability

The datasets generated during the current study are available from the corresponding author on reasonable request. Molecular data underlying the findings described in the manuscript are fully available without restriction from the Bioproject Sequence Read Archive (https://www.ncbi.nlm.nih.gov/sra/PRJNA833243; http://www.ncbi.nlm.nih.gov/bioproject/PRJNA613395; https://www.ncbi.nlm.nih.gov/bioproject/PRJNA1073422).

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
