# Peer review of "The C250T Mutation of *TERTp* Might Grant a Better Prognosis to Glioblastoma by Exerting Less Biological Effect on Telomeres and Chromosomes Than the C228T Mutation"

_cancers, 2024, doi:10.3390/cancers16040735_

Round 1

Reviewer 1 Report

Comments and Suggestions for Authors

Dear Editor,
Dear authors,

In this article, Gorria et al. analyzed the correlation between hTERTp mutations C250T/C228T and other prognostic factors in a series of glioblastoma patients. They concluded that C250T was linked to a better prognosis. The manuscript is well writing, the aim of the study is clear and well defined. These data will help future analyses to understand the complex mechanism of TERT (and telomerase) regulation and implication in cancers. Please find below some comments and suggestions:

Minor comments:

-      In the introduction section, I suggest moving the 4th paragraph (line 8-93) to the 2nd place (line61). I guess it would be clearer to introduce the telomeres, telomerase and TERTp mutations before giving more details.

-      I suggest adding the proportion of cancers activating the telomerase (85-90%) and those who are ALT positive (PMID: 30760854)

-     Also, line 89, I guess it is worth mentioning that TERT genetic and epigenetic alterations are reported in various neoplasms leading to Telomerase activation (PMID: 33329574, 33715271, 34720085)

-      In the 2.2 section, line 120-122: what does it mean that “The lack of detection of TERTp mutations on these last two tests did not automatically classify the tumor as TERTp-wt.” ? please clarify this idea.

-         In the legend of table Table 1 :  the explanations of  “a” and “b” permuted. Please correct.

-          Line 258 and 260: I suggest using the word “Variants” instead of “mutations”, since as mentioned in line 261 one of these variants is benign. Also, I suggest using “Pathogenic” instead of “Harmful.

-          Table 3 and line 270: what if the analysis was done based on C250 vs wt+C228T? Do we observe any significant P value? Especially for the borderline value observed in BRAF (P=0.052)

-          At the end of the results section, end of last paragraph: we feel that the last sentence presents some sort of interpretation rather than just a result.

-          How was the evaluation of TERT expression done (line 363-364)? It is not clearly mentioned in the Materials section. Was TERT expression done for all samples included?

-          In the limitations, and absence of correlation between TERT expression level and TERTp mutations, several factors can play a role including promoter methylation, and maybe also the TERT transcripts (Alpha and Beta, combinations). These different transcripts play different roles and are present in different proportions in several cancers, thus impacting the role of TERT expression (PMID: 17130181, 36833366)

-          TERTp mutations loci are not covered by WES? I guess it would be worth mentioning in the manuscript.

Comments on the Quality of English Language

No comments.

Reviewer 2 Report

Comments and Suggestions for Authors

I absolutely love this paper. The study addresses an important question, is well powered, and determined relevant molecular and clinical variables in the analysis. The manuscript is well written.

I only have two minor suggestions. First, in Supplementary Fig 4, p values were given for each pairwise comparison. However, "KW, p=0.25" in the figure seems to be out of the place. It is unclear what this p value is for, and there is no legend to explain.

The second comment is about the title of this paper. The study is very strong in survival analysis, cohort size, etc., but the mechanistic insights on why 228T mutations are associated with inferior outcomes are mainly from RNAseq data. Simple comparison of gene and pathway expression does not convey very strong mechanistic data. Regretably the authors chose to highlight this in the title, but the title is clearly speculative, suggesting the authors themselves have concerns. I'd suggest the authors to revise the title, but this would be up to them. 
